# Comment on Ambra et al. Could Selenium Supplementation Prevent COVID-19? A Comprehensive Review of Available Studies. *Molecules* 2023, *28*, 4130

**DOI:** 10.3390/molecules29112466

**Published:** 2024-05-24

**Authors:** Margaret P. Rayman, Lutz Schomburg, Jinsong Zhang, Ethan Will Taylor, Gijs Du Laing, Melinda Beck, David J. Hughes, Raban Heller

**Affiliations:** 1Department of Nutritional Sciences, University of Surrey, Guildford GU2 7XH, UK; 2Institute of Experimental Endocrinology, Charité—Universitätsmedizin, D-10115 Berlin, Germany; lutz.schomburg@charite.de (L.S.);; 3State Key Laboratory of Tea Plant Biology and Utilization, School of Tea & Food Science, Anhui Agricultural University, Hefei 230036, China; 4Department of Chemistry and Biochemistry, University of North Carolina at Greensboro, Greensboro, NC 27402, USA; ewtaylor@uncg.edu; 5Faculty of Bioscience Engineering, Ghent University, 9000 Ghent, Belgium; 6Gillings School of Global Public Health, University of North Carolina at Chapel Hill, Chapel Hill, NC 27599, USA; melinda_beck@med.unc.edu; 7School of Biomolecular and Biomedical Science, UCD Conway Institute, University College Dublin, D04 V1W8 Dublin, Ireland

The authors of this Comment are longstanding selenium investigators with a total of 200 or more published articles on selenium; the corresponding author (Margaret P. Rayman) has published two highly cited reviews on selenium and human health in *The Lancet* (2000 and 2012), and Lutz Schomburg and his team are highly respected as selenium researchers worldwide. We noticed that a review paper by Ambra et al. entitled “*Could Selenium Supplementation Prevent COVID-19? A Comprehensive Review of Available Studies*” [1] was recently published in *Molecules* in a Special Issue on “Functional Foods and Dietary Bioactives in Human Health”, in the Section “Natural Products Chemistry”.

Ambra et al. [1] cite and discuss more than 10 papers by the authors of this Comment. We take exception to most of their assessments of our work, but due to space limitations will only give detailed rebuttals for two of them [2,3], as these cases are illustrative of the inaccuracies and misinterpretations that pervade the whole review.

In the Introduction, it already seems that the role of selenium has been misrepresented—it does not modulate the biochemical activity of several enzymes; indeed, enzymes containing selenocysteine are the *key* factors. Furthermore, the effects of selenium on mortality are not clearly described; selenium status varies across the world and is linked in a U-shape with selenium mortality.

First, the article is *not* a comprehensive review of available studies, despite that word being highlighted in the title. At least two important studies published prior to the submission date were not included, i.e., “Association of COVID-19 mortality with serum selenium, zinc and copper: Six observational studies across Europe” [4] and “COVID-19 Severity Is Associated with Selenium Intake among Young Adults with Low Selenium and Zinc Intake in North Carolina” published by American Society for Nutrition [5]. Both of these studies used rigorous methods for the determination of selenium status (either in serum or via toenail analysis) in cohorts of actual COVID-19 patients and found that selenium had significant correlations with COVID-19 outcomes.

It was quite surprising that they dismissed on our work [2] without mentioning our major findings while only discussing its possible biases (which we ourselves had listed as limitations). We now wish to clarify their mistaken comments. As China has populations that have both the lowest and the highest selenium status in the world, we examined cumulative data on the cure rate of COVID-19 in Chinese cities with more than 40 cases (18 February 2020). Regarding selenium status, we found that 17 cities outside Hubei (the epicentre province in China) had documented data on hair selenium. We observed a significant association between cure rate and background hair selenium concentration (R^2^ = 0.72, F_test_
*p* < 0.0001). We clearly outlined that our study had several limitations, such as non-current selenium status, likely confounders including age and comorbidities, a lack of information on variation in medical facilities and therapy protocols (including the use of traditional Chinese medicine or anti-viral therapies). Therefore, we were not able to adjust for these possible confounders in the analysis, particularly as data on these variables (age, comorbidities, etc.) were not included in the data set available for our analysis. We are fully aware, therefore, that the association shown is far from being robust to criticisms of confounding. At best, it points towards the need for further research. Surprisingly, without introducing our main finding (the association of cure rate and background hair selenium), they took issue with our report.

Ambra et al. criticise (in reference to our article) that “the authors stated that the selenium status was available only for two cities inside the Hubei Province; however, selenium status is mentioned only for the city of Enshi“. This hardly constitutes “bias”, especially as we clearly stated that the Enshi (high-selenium) data should be compared to “typical levels in Hubei of 0.55 mg/kg” [6]. In Table S2 of our paper [2], there are two hair selenium reports concerning Hubei province. In each report, hair selenium samples were collected from inhabitants of Wuhan city (the capital of Hubei province and the second city mentioned in their question). The hair selenium level of 0.55 mg/kg was taken from the two reports; they have been weighted by numbers of samples per study. In any case, their concern is not warranted, as the important point is that there were obviously insufficient selenium data from cities inside Hubei to allow a multi-city correlation to be made, as we did for the cities outside Hubei.

Ambra et al. further state, “Moreover, even if the city of Enshi (death rate 1.6) is reported for being an exception to the high mortality rate of Hubei Province (high death rate) based on the data shown, other cities (Shiyan and Xianning) inside Hubei province have a lower death rate (0.3 and 1.2)”. This criticism ignores our key evident point that Enshi had by far the highest *cure* rate (36%) of *any* city in Hubei, including both Shiyan (23%) and Xianning (24%). In the correlation made for cities outside Hubei (our Figure 1 [2]), we also focused on the cure rate as opposed to the death rate, because so early in the pandemic, many cases were still unresolved (neither cure nor death) at those early pandemic stages, thus a low initial death rate early on might soon be reversed. In contrast, a high cure rate early on, even with many unresolved cases, becomes even more significant. Regarding Shiyan, in the top quartile for cure rates for cities in Hubei (our Table S1 [2]), it is also notable that in a later study, Liu et al. identified it as a “selenium-enriched city”, based on soil selenium levels [7]. The ensuing comments of Ambra et al. then become difficult to understand as they mention newer data on the “incidence of diagnosed people by April 30”. Aside from our study being based on a snapshot of the outbreak on 18 February 2020, it did not even attempt to examine any links between Se status and the city-associated *incidence* of COVID-19 cases (see the penultimate paragraph).

Results published by Lutz Schomburg’s group (Ambra references 59, 60, 61, 69) were either erroneously criticised or ignored. Furthermore, the authors of the review are clearly unaware of the remarkable results reported by Melinda Beck, a distinguished scientist and a major pioneer in selenium and virus research. They even stated that Beck’s approach, which used serum glutathione peroxidase—which dropped from 33.0 ± 4.4 to 4.7 ± 0.2 munits/mg protein—to demonstrate selenium status, was unconvincing.

Ambra et al. stated that some conclusions drawn by Du Laing et al. [3] were not supported by the statistical analysis. However, Ambra et al. focused only on the absence or presence of statistical significance at the 0.05 level (as presented in bold in Table 2), whereas Du Laing et al. also took the *p*-values, as well as the differences in means between the groups, into account when evaluating effects and differences between the groups. The statistical methods used by Du Laing et al. were very conservative, suited to the limited numbers of subjects within the subgroups.

The absence of data on the length of hospital stays and information on vaccination status is also criticized by Ambra et al., but these data were available upon request, as stated in the Data Availability Statement at the end of the paper. However, Ambra et al. never requested these data from the corresponding author. Confusion on vaccination status is attributed by Ambra et al. to ethics having been approved in March 2021, which is incorrect; Du Laing et al. clearly stated that ethics approval was given in March 2020 (see the Institutional Review Board Statement included in the paper).

Ambra et al. furthermore stated, “All non-survivors displayed a strong selenium deficit ([Se] < 55.2 µg/L) already at hospital admission is denied by Figure 2, which shows that at least two of the patients who died at the end of the study had, at admission, a selenium concentration over this amount”. However, Du Laing et al. clearly stated that all non-survivors displayed selenium levels below the threshold for moderate Se deficiency (<70 µg/L), so not 55.2 µg/L. Furthermore, Ambra et al. subsequently claim that the selenium thresholds used in the paper were unclear. Du Laing et al. did indeed use several thresholds in the first part of the text, including existing thresholds reported in the literature, but afterwards selected a new threshold when developing their own model (presented in their Table 4). However, the thresholds used when evaluating selenium or zinc deficiency/sufficiency were always clarified, as can be seen in the text, footnotes, and captions of the tables and figures in the Du Laing paper.

Ambra et al. criticized the fact that Du Laing et al. reported 10 deaths out of 73 patients “instead of 79”. However, the text explained that five patients from the 79 enrolled in study 1 were transferred to another facility, and one patient was still hospitalized at the end of the study period. These six patients were not included when evaluating mortality.

Perhaps we should be asking whether the question the authors asked—*Could Selenium Supplementation Prevent COVID-19?*—is even a reasonable question. It does not adequately reflect our understanding of the effect of selenium on viruses. On the assumption that Ambra et al. mean COVID-19 *infection* in their title, we acknowledge that selenium supplementation does not have a sufficient direct antiviral effect to prevent infection, but tends to act by decreasing the harmful effects of the virus on the host. So it does not necessarily *prevent* infection but rather decreases the severity and mortality resulting from the infection. The expectation that selenium must have antiviral and preventive effects risks throwing out the baby with the bath water, because it does not place sufficient value upon virus–host interaction as the cause of pathogenic effects. In that regard, it is important to put the clinical correlations between dietary selenium status and COVID-19 outcomes in the context of evidence regarding the effects of SARS-CoV-2 infection on host selenoprotein status and mechanisms, which include disruption to the biosynthesis and endogenous levels of certain selenoproteins in the host [8,9,10]. Such results suggest that the importance of Se in COVID-19 is not due simply to effects on overall selenoprotein expression, but that the virus is targeting the unique biochemistry of specific selenoproteins, independently of the dietary hierarchy. This highlights the urgency of continued investigation into the roles of selenium in COVID-19 and the irresponsibility of a cavalier dismissal of the evidence for its importance.

We have commented on this review because it purports to assess the whole area of selenium and COVID-19 and because not to do so would allow readers to maintain a misunderstanding of the validity of much of the original, robust data provided by the current authors.

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
