# Peer review of "Comment on Ambra et al. Could Selenium Supplementation Prevent COVID-19? A Comprehensive Review of Available Studies. Molecules 2023, 28, 4130"

_molecules, 2024, doi:10.3390/molecules29112466_

Round 1

Reviewer 1 Report

Comments and Suggestions for Authors

The manuscript is raised by one of the experts for selenium epidemiology. The discussion contains many merits for the audience.

Author Response

Thank you for your supportive comments.

Reviewer 2 Report

Comments and Suggestions for Authors

I completely agree that the current title "Could selenium supplementation prevent COVID-19?" is not appropriate since selenium (Se) is an essential trace nutrient rather than a medicine. In this case, I am not surprised by the conclusions presented by Ambra et al. (2023), which state that Se supplementation did not prevent SARS-CoV-2 infection or improve COVID-19 prognosis. However, I find the criticism style used by Ambra et al. to criticize all research on Se here unacceptable, as it seems that Se research on COVID-19 is nonsensical. This criticism, which is misleading to the general public and researchers outside the field of Se research, especially young researchers, is not acceptable. It may also undermine confidence in Se among young scientists conducting Se research in the future.

Furthermore, the present review paper intentionally overlooked or disregarded relevant and important publications, such as "Association of COVID-19 mortality with serum selenium, zinc, and copper: Six observational studies across Europe. Frontiers in Immunology, 2022, 13: 1022673." and "COVID-19 severity is associated with selenium intake among young adults with low selenium and zinc intake in North Carolina. 2023, 7(3): 100044." This omission or ignorance of important studies weakens the overall validity of the review paper.

Therefore, I support the publication of the comments from Rayman et al., as they will provide a more comprehensive understanding of Se for the readers.

Author Response

I completely agree that the current title "Could selenium supplementation prevent COVID-19?" is not appropriate since selenium (Se) is an essential trace nutrient rather than a medicine. In this case, I am not surprised by the conclusions presented by Ambra et al. (2023), which state that Se supplementation did not prevent SARS-CoV-2 infection or improve COVID-19 prognosis. However, I find the criticism style used by Ambra et al. to criticize all research on Se here unacceptable, as it seems that Se research on COVID-19 is nonsensical. This criticism, which is misleading to the general public and researchers outside the field of Se research, especially young researchers, is not acceptable. It may also undermine confidence in Se among young scientists conducting Se research in the future.

Furthermore, the present review paper intentionally overlooked or disregarded relevant and important publications, such as "Association of COVID-19 mortality with serum selenium, zinc, and copper: Six observational studies across Europe. Frontiers in Immunology, 2022, 13: 1022673." and "COVID-19 severity is associated with selenium intake among young adults with low selenium and zinc intake in North Carolina. 2023, 7(3): 100044." This omission or ignorance of important studies weakens the overall validity of the review paper.

Therefore, I support the publication of the comments from Rayman et al., as they will provide a more comprehensive understanding of Se for the readers.

Thank you for your observations and your opinion that our Comment will provide a more comprehensive understanding of Se for the readers.

Reviewer 3 Report

Comments and Suggestions for Authors

In my opinion, the work should be published. The authors have clarified all the doubts in the review.

Author Response

Thank you for confirming that our Comment should be published as it has clarified all the doubts in the Ambra et al. review.

Reviewer 4 Report

Comments and Suggestions for Authors

My comments and suggestions of Authors of both Comment and Reply would be to attenuate their comments and potential remarks on others' work. I believe that more good for scientific community can be achieved through constructive criticism rather through snarky remarks and "pulling out" a conclusions which have no real support in written words. 

Author Response

We have attenuated some of our remarks on the Review of Ambra et al. in line with Reviewer 4’s request.

Reviewer 5 Report

Comments and Suggestions for Authors

The authors of this comment wrote an answer to the review article of Ambra et al. published in Molecules 2023; 28(10). They disagree with many interpretations of Ambra et al. concerning the studies included in their review article. They also mention that two important studies were not included in this review article, which is true.

I agree with the comment, which I believe is important to publish as a contrast to the review article of Ambra et al. I have read and compared the points brought up by the authors of the comment and agree with them.

There is one point that I would like to mention. In the second last paragraph of the comment, the authors state “Perhaps we should be asking whether the question the authors asked - Could Selenium Supplementation Prevent COVID-19? – is a reasonable question.” In the following, the authors answer this question by assuming that Ambra et al. meant “COVID-19 infection” by “COVID-19” in their question Could Selenium Supplementation Prevent COVID-19? However, COVID-19 could also mean COVID-19 illness. So, I would be more prudent regarding the statement and start with something like “If Ambra et al mean COVID-19 infection in their question asked …

Furthermore, on line 68 it should mean “lack of information” or “lacking information”.

Author Response

The authors of this comment wrote an answer to the review article of Ambra et al. published in Molecules 2023; 28(10). They disagree with many interpretations of Ambra et al. concerning the studies included in their review article. They also mention that two important studies were not included in this review article, which is true.

I agree with the comment, which I believe is important to publish as a contrast to the review article of Ambra et al. I have read and compared the points brought up by the authors of the comment and agree with them.

There is one point that I would like to mention. In the second last paragraph of the comment, the authors state “Perhaps we should be asking whether the question the authors asked - Could Selenium Supplementation Prevent COVID-19? – is a reasonable question.” In the following, the authors answer this question by assuming that Ambra et al. meant “COVID-19 infection” by “COVID-19” in their question Could Selenium Supplementation Prevent COVID-19? However, COVID-19 could also mean COVID-19 illness. So, I would be more prudent regarding the statement and start with something like “If Ambra et al mean COVID-19 infection in their question asked …

Thank you for agreeing with our Comment and expecting it to be published as a contrast to the review article of Ambra et al.

With regard to your point as to whether Ambra et al mean COVID-19 infection in their question, we have changed our Comment to clarify that.

Furthermore, on line 68 it should mean “lack of information” or “lacking information”.

Amended to read “lack of information”.